# Development and In-Field Validation of an Autonomous Soil Mechanical Resistance Sensor

**DOI:** 10.3390/s25061919

**Published:** 2025-03-19

**Authors:** Valentijn De Cauwer, Simon Cool, Axel Willekens, Sébastien Temmerman, David Nuyttens, Tommy D’ Hose, Jan Pieters, Sam Leroux

**Affiliations:** 1Technology and Food Science Unit, Flanders Research Institute for Agriculture, Fisheries and Food (ILVO), 9820 Merelbeke-Melle, Belgium; simon.cool@ilvo.vlaanderen.be (S.C.); axel.willekens@ilvo.vlaanderen.be (A.W.); sebastien.temmerman@ilvo.vlaanderen.be (S.T.); david.nuyttens@ilvo.vlaanderen.be (D.N.); 2Department of Plants and Crops, Ghent University, 9000 Ghent, Belgium; jan.pieters@ugent.be; 3Plant Science Unit, Flanders Research Institute for Agriculture, Fisheries and Food (ILVO), 9820 Merelbeke-Melle, Belgium; tommy.dhose@ilvo.vlaanderen.be; 4Department of Information Technology, Ghent University, 9000 Ghent, Belgium; sam.leroux@ugent.be

**Keywords:** soil compaction, automated penetrometer, soil penetration resistance, mechanical resistance, autonomous robot, precision agriculture, validation, spraying tracks

## Abstract

Soil compaction is a widespread problem, leading to soil degradation, yield losses, and adverse environmental impacts. Nowadays, various measurement methods exist to assess and map soil compaction, with vertical cone penetration resistance measurements being one of the most commonly used. This method is easy, rapid, inexpensive, and generally accepted. However, manual penetration resistance measurements are time-consuming, labor-intensive, and often less accurate due to inconsistent penetration speed. To address these limitations, an automated penetrometer was developed and integrated on an autonomous robot platform, paving the way for high-resolution compaction mapping as a starting point for precision subsoiling to remediate soil compaction. The performance of this setup was validated in controlled and field conditions against a hand-held penetrometer. Therefore, experiments were conducted in soil-filled cylinders and on plots of a long-term field experiment, including measurements across spraying tracks. The automated penetrometer demonstrated high correlations with the hand-held device under controlled conditions, though the correlation was somewhat lower in the field due to the soil’s heterogeneity. Deviations between the two measurement devices were likely caused by the inconsistent insertion speed of the hand-held penetrometer, particularly in soils with high penetration resistance. Both penetrometers successfully identified the plow pan at a depth of 30–40 cm but were unable to clearly show the effect of the long-term presence of spraying tracks.

## 1. Introduction

Today’s challenge of providing a growing world population with sufficient food in a changing climatic environment is greater than ever. The world population is expected to grow to approximately 9.7 billion by 2050, accompanied by an increasing food demand [1,2]. As a consequence, there is an urgent need to continuously improve the agrifood system, with a particular emphasis on plant-based production [3]. Where possible, manual labor is replaced with cost-effective machinery [4]. To expand capacity, agricultural machines have become increasingly wider and heavier, leading to higher wheel loads [5,6,7]. Schjønning et al. [5] further noted that, despite modern tires being more voluminous, the expansion of the tire–soil contact area has not kept pace with the increase in wheel load, leading to higher ground pressures. Consequently, the rising wheel loads have aggravated the level of soil compaction in agricultural soils, particularly in the subsoil [4,6,8,9,10]. Subsoil compaction is defined as compaction that regular tillage practices cannot alleviate, as it occurs below the annual tillage depth; hence, it is considered the most problematic form of compaction [10]. Unfavorable weather conditions, especially prolonged wet periods, further increase the vulnerability of arable land to (sub)soil compaction, thereby increasing both the risk and prevalence [11,12]. The projected increase in extreme weather events might therefore exacerbate the problem of soil compaction. In Belgium, the harvest of crops such as (silage) maize, sugar beets, and potatoes is often associated with high axle loads, together with moist soil conditions in late summer/autumn. In addition, slurry and manure applications in early spring are often accompanied by high soil moisture levels, posing a high risk of soil compaction.

Nongovernmental organizations and green political movements pay little attention to the soil compaction problem, as its effects are not always directly visible [5]. However, the consequences of subsoil compaction are far-reaching, affecting almost every soil function [13]. The most important consequence of (subsoil) compaction is the increased bulk density [14,15,16,17]. Other changes in soil physical properties, such as total soil porosity and pore system morphology, may also occur [5,18]. Due to poor aeration, higher denitrification rates and lower mineralization rates were observed [19], resulting in soil nitrogen loss and the potential to enhance global warming through the production of N_2_O. Furthermore, compaction impairs water flow, reduces infiltration, and intensifies runoff and erosion [14,16,17,20]. It also significantly affects agricultural productivity, restricting root growth, nutrient uptake, root depth, root aeration, and water uptake [5,21,22,23]. As a result, numerous studies showed yield reductions for various crops due to compaction [17,21,22,24,25,26,27,28]. Because subsoil compaction occurs below the regular tillage depth, detecting and remediating it is challenging and often requires deep tillage operations (such as subsoiling) and/or crop rotations with deep-rooted plants [29,30]. Deep tillage, however, is a highly energy-intensive tillage operation [31,32], and it often fails to restore the soil structure to that of non-compacted soil [33,34]. Furthermore, there is a risk of recompaction after subsoiling, especially if not followed by soil conservation measures [35,36,37,38].

Quantifying the area affected by soil compaction and determining the severity is challenging and time-consuming [5,39]. Around 68.3 million hectares are globally affected, with Europe accounting for 33.0 million hectares [40,41]. Little is known about the extent of soil compaction in Belgium. Vanderhasselt [42] revealed, based on a field study of 42 fields in the western part of Belgium, that between 27% and 87% of the subsoil was critically compacted, depending on the soil quality parameter considered. Van De Vreken et al. [43] also provided a soil compaction susceptibility map for Belgium. Given the widespread occurrence and serious consequences of (sub)soil compaction, effective monitoring and field mapping are crucial [6,43] to (1) manage land degradation [44], (2) estimate the severity of compaction [4], (3) develop and refine sustainable management guidelines [4], and (4) explain crop yield variations within a field [45,46]. Furthermore, no rules regarding soil compaction are applicable in the European Union (EU), leaving farmers reliant on their own judgment and advisor guidelines, underscoring the need for soil compaction data [47]. Additionally, creating digital compaction maps could enable variable depth tillage to remediate compacted layers efficiently without wasting resources, such as fuel and labor, and limiting soil disturbance to where it is necessary [48,49,50].

Keller et al. [4] reviewed techniques for quantifying and mapping soil compaction, including in situ and non-invasive methods. The EU-funded ENVASSO project identified bulk density and air-filled pore space at a predefined matric head as key indicators [8]. Other relevant parameters include pore space, void ratio, soil depth (i.e., the volume of soil in which roots can grow), packing density, water retention properties, soil erosion rate, and surface sealing [51,52]. However, the previous indicators have some disadvantages. In general, traditional measurements based on soil sampling are invasive, disturb the soil, and require laboratory analysis, making them time- and resource-intensive [6,8,53,54]. To address these challenges, several geophysical techniques have been explored, including direct current resistivity, electrical resistivity tomography (ERT), electromagnetic induction (EMI), induced polarization (IP), and ground-penetrating radar (GPR), with varying degrees of success [55,56,57,58,59,60,61,62]. However, these methods often fail to provide accurate profile information, and their results can be difficult to interpret. In contrast, horizontal penetrometers have been developed, which allow for continuous mechanical resistance measurements at specific depth intervals [63,64,65]. Other sensors that are dragged through the soil have also been designed, including implements that measure the bending moment of wings attached to a tine or the draft force of a tine [50,64,66,67].

Despite these advancements, the vertical penetrometer remains widely used for assessing soil compaction at continuous depths [8,68,69,70,71]. Studies have confirmed the link between compaction, bulk density, and cone resistance [16,22,27,72]. Measuring vertical cone resistance is fast, simple, and cost-effective [73], but dependent on soil moisture levels and cone specifications [8]. De Moraes et al. [71] emphasized the need for standardization in penetration resistance measurements for proper comparison with literature values. They acknowledge that measuring penetration resistance at field capacity is considered good practice. Efforts to establish standardization include the ASABE S313.3 standard [74]., which defines specifications for penetrometers, categorizing them into soft and hard soil devices with maximum resistance levels of 2 MPa and 5 MPa, respectively. Specifications include the enclosed angle and the base area of the penetration cone. Additionally, the ASABE prescribes a fixed insertion velocity of 30 mm/s [71,74]. According to Mahore et al. [75], manual penetrometers use sensors to measure penetration depth, which require regular calibration. They are manually inserted into the soil at an inconsistent rate, negatively affecting accuracy, and often offer limited real-time accessibility. Additionally, operating these devices can be time-consuming and difficult (e.g., hard soils) [73]. To address these issues, researchers developed hydraulically driven tractor-mounted penetrometers that ensure a fixed insertion velocity, enabling faster and more accurate measurements [76,77,78,79]. Furthermore, Mahore et al. [75] introduced a portable, constant-rate, IoT-enabled cone penetrometer for real-time data collection.

Some authors argue that on-the-go sensors are needed to map soil compaction at a useful spatial resolution [4]. An alternative approach is a robot-mounted automated penetrometer, which allows for time-efficient measurements and additional data collection while maintaining comparability with manual penetrometers. Fentanes et al. [80] and Scholz et al. [81] developed such systems. However, Fentanes’ design is limited to a maximum measuring depth of 50 cm and lacks a side shift, while Scholz’s version includes a side shift but allows only four replicates. Additionally, both penetrometers require a specific attachment method rather than a standard three-point hitch.

Our automated penetrometer overcomes these limitations and integrates with the open-source Agricultural Robot Task map Operation Framework (ARTOF) developed by Willekens et al. [82] (under review), making it compatible with a wide range of autonomous robots. This paper aims to describe the in-house developed, autonomous robot-mounted automated penetrometer for high-resolution soil compaction mapping and validate this penetrometer under both controlled and field conditions by comparing it with manual penetrometer measurements.

## 2. Materials and Methods

### 2.1. Integration with Autonomous Robot Platform

The automated penetrometer has been integrated into the rear three-point hitch of the in-house developed CIMAT robot. This robot was developed during the European Interreg Flanders-Netherlands project named Catalyst for Innovative Mechatronics in Agricultural Technology (CIMAT). The technical specifications of the CIMAT robot were described by [82] (under review). In short, the CIMAT robot is a four-wheel-drive, four-wheel-steer (4WD4WS) electric-powered robot that was mainly developed for crop-care applications. The mean power consumption of the robot when performing penetrometer measurements is approximately 2.36 kW, which means that, with a battery capacity of 20 kWh, the robot can operate continuously for an estimated duration of more than 8 h. The open-source Agricultural Robot Task map Operation Framework (This framework is available at https://github.com/artof-ilvo (accessed on 9 December 2024) under the ILVO License, with documentation accessible at https://artof-ilvo.github.io. (accessed on 9 December 2024)) (ARTOF) developed by Willekens et al. [82] (under review) allows the robot to perform applications according to a task map. A task map comprises a trajectory, geofence, and tasks defined using ESRI shapefiles (see Section 2.2.4). The tasks are categorized as continuous, discrete, and intermittent. The discrete operations functionality was developed specifically for soil pressure sampling. The robot halts along the trajectory to take a discrete measurement when the implement reference is closest to a measurement location. The implements can communicate with the robot platform using the Redis ARTOF interface or by Siemens^®^ S7 communication (Siemens AG, Munich, Germany). A Node-red^®^ add-on of the ARTOF can be used to add implement-specific functionality (see Section 2.2.3).

### 2.2. Automated Penetrometer

#### 2.2.1. Hardware

The automated penetrometer consists of a probe, which includes a rod and penetration cone, a load cell (T60-100KG, Thames Side, Reading, United Kingdom), a PLC (SIMATIC S7-1200, Siemens AG, Munich, Germany), and linear actuators driven by stepper motors with absolute encoders (Figure 1). The penetration rod is 97 cm long and has a diameter of 0.8 cm. Since the ASABE standard is not widely accepted and to ensure comparability with the recommended cone for general fieldwork with the hand-held Eijkelkamp penetrometer (06.15.SA, Giesbeek, The Netherlands), which is commonly used in Belgium, a penetration cone with a 60° angle and a base area of 1 cm^2^ was chosen. The penetration probe is attached to a vertical linear actuator (MTV65, Unimotion, Lesce, Slovenia), powered by a stepper motor (AZM69MK, Oriental Motor GmbH, Düsseldorf, Germany), and controlled by a motor driver (Oriental Motor, AZD-KD). The vertical system can be displaced horizontally (side shift) along a double guide rail by a horizontal linear actuator (Unimotion, CTV110), which is powered by a stepper motor (Oriental Motor, AZM66AK) and controlled by the same motor driver used for the vertical linear actuator. This side shift enables the collection of seven horizontal measurement replicates, typically spaced 10 cm apart. A series of horizontal point measurements is called a measurement series. A second vertical linear actuator is available for the future integration of a soil moisture sensor.

#### 2.2.2. Programmable Logic Controller Program

The automated penetrometer program is implemented on the penetrometer’s PLC. Figure 2 illustrates the procedure for a complete penetration resistance measurement series down to a depth of 75 cm. The procedure includes three phases: (1) initialization, (2) execution of a measurement series, and (3) execution of a single point measurement. First, the measurement procedure is initialized by lowering the three-point hitch until the penetrometer’s landing legs reach the ground surface, moving the linear actuators to their home position, and disabling the traction drives. Next, a measurement series is initiated by performing the first point measurement. To conduct a point measurement, the penetrometer’s cone tip moves vertically to the ground position (6550 steps = 3.275 cm) at a speed of 15 cm/s. The ground position is at the same height as the lower part of the landing legs. After reaching the ground position, a single point measurement is conducted. The penetrometer rod is inserted into the soil at a velocity of 2 cm/s, following the recommended operation speed of the hand-held Eijkelkamp penetrometer, as specified in the device’s manual [83]. Every 0.5 cm, the load cell records a force measurement, which is saved on an SD card. This process continues until the rod reaches the vertical end position (ground position + 75 cm) or the measured force exceeds a preset maximum value of 80 kg (=784.9 N). This maximum value is set to prevent damage to the penetrometer, as it represents the mechanical limit of the vertical linear actuator. Once the rod stops, it returns to its home position (0.5 cm) at a speed of 10 cm/s. After each point measurement, the horizontal linear actuator shifts the vertical actuator by 10 cm (adjustable), until the PLC confirms that all point measurements in the measurement series are completed. At this point, the horizontal linear actuator moves to the home position (70 cm) at a speed of 15 cm/s. Finally, the three-point hitch is lifted, and the traction drives are enabled to move to the next RTK-GPS location.

#### 2.2.3. Data Registry

A Node-RED flow (Accessible at https://flows.nodered.org/flow/7f8c086d8ba28efbd663e6cb51565920 (accessed on 23 January 2025)) for the automated penetrometer is integrated into the CIMAT robot’s Node-RED add-on (Appendix A). Each time a point measurement is completed, Node-RED extracts an array with pressure values (MPa), depths (mm), horizontal positions (cm), and times elapsed since the start of the measurement (ms) from the penetrometer’s PLC using Siemens S7 communication. The Node-RED flow also collects robot state information—including the RTK-GNSS position from the section reference (Figure 3) and the vertical position of the three-point hitch—from the ARTOF Redis interface. By using the transformation chain from the hitch hinge to the reference section (as defined in the implement configuration file) and taking into account both the hitch position relative to the robot and the horizontal offset of the rod, Node-RED calculates the exact position of the rod during a point measurement. Additionally, Node-RED calculates the datetime for each datapoint of the extracted array by subtracting the total elapsed time from the current datetime and then adding the elapsed time for each individual datapoint. Finally, Node-RED combines the data and sends this to the time-series database instance of InfluxDB. Optionally, the data of this database can be visualized in an interactive Grafana dashboard.

#### 2.2.4. Task Map Execution

To execute a task map, a trajectory, geofence, and discrete task map are created as an ESRI shapefile in QGIS (Version 3.34.10 Long Term Release (download from https://qgis.org/download/ (accessed on 20 September 2024)) and are loaded onto the CIMAT robot. Each measurement series is initiated via the ARTOF and performed at a specific measurement location defined by an RTK-GNSS coordinate. The trajectory defines the robot’s driving direction and the sequence in which the task map points will be completed. The task map consists of RTK-GNSS coordinates, which precisely define the measurement locations. The robot halts along the trajectory to take discrete measurements when the implement section reference matches the measurement location. The ARTOF keeps track of the penetrometer’s section reference position during operation, enabling the robot to slow down when the penetrometer approaches a measurement location. When the section reference passes this point, a stop command is immediately initiated, the three-point hitch is lowered, and the measurement series is initiated.

### 2.3. Calibration and Validation Under Controlled Conditions

A validation between the developed automated penetrometer and a hand-held commercially available penetrologger (Eijkelkamp, 06.15.SA) was performed under controlled conditions. The hand-held device includes a built-in datalogger and an integrated GPS, and it records the penetration resistance at every 1 cm depth interval, up to a maximum depth of 80 cm. The rod and cone specifications of the hand-held penetrometer are identical to those of the automated penetrometer.

Before validation, the calibration of the load cells of both the hand-held and the automated device was verified using a spring, a metal piece that fit into the opening of the spring, and a balance (Figure 4a). The metal piece had a small indentation in which the penetration cone could be stabilized when pushing the rod down. For the calibration of the hand-held device, a clamp was constructed to mount it onto the robot’s linear actuator (Figure 4b, hereafter referred to as the robot-mounted hand-held penetrometer). Additionally, an extra bearing was included to guide the rod. The frictional pressure from this bearing was always below 0.03 MPa, indicating that it did not introduce any significant additional resistance. The spring was used solely to allow the rod to move over a distance without damaging the penetrometer. The spring was placed in an iron casing, to ensure a safe and stable working environment, and positioned on top of the balance. The robot’s PLC was programmed to enable the gradual manual lowering of the penetration rod through the robot’s operation screen to obtain readings at different forces. The data from both penetrometer devices and the balance were recorded manually.

The validation conducted under controlled conditions involved point measurements within steel cylinders filled with soil. Cylinders, each 75 cm in length with an inner diameter of 10.5 cm, were filled with 8.48 kg of sandy loam soil (at field capacity). Before filling the cylinders, the soil was homogenized by mixing and crumbling. The soil was then compressed to a depth of 15.5 cm from the cylinder’s top. Next, the cylinder was placed vertically and stabilized in a rain pipe that was buried in the soil (Figure 5). This procedure allowed for multiple replicate measurements in a standardized manner. Each time, a measurement was performed at the center of the cylinder. Ten replicate measurements were performed for each of the three measurement methods: the hand-held penetrometer (each time the same operator), the robot-mounted hand-held penetrometer, and the automated penetrometer. Measurements with the first two methods were performed on the same day, while the third method was tested two days later. In the meantime, the preprocessed soil was stored in plastic bags. Since both penetrometers could not detect the soil surface in the cylinders, the measurement data were manually shifted to ensure the pressure was zero at the soil surface.

### 2.4. Field Validation

For the field validation, we built upon an existing multi-year field trial. The experimental field (50°59′6.84″ N, 3°46′24.10″ E) is located at the ILVO test site in Merelbeke-Melle, Belgium. Due to the field’s topography the northeast side generally has a higher soil moisture content compared to the southwest side. According to the Belgian soil map, the field is classified as light sandy loam soil with a strongly mottled, broken-texture B horizon. The field experiment investigates the effect of non-inversion tillage versus conventional plowing, as well as different organic fertilizers, on various soil parameters. For further reading, we refer to the paper of D’Hose et al. [84]. An overview of the experimental field design is provided in Figure 6. The main crop grown in the year of our research was silage maize.

For the experimental design of the field validation, measurements were conducted on the experimental plots using the automated penetrometer in May 2024 (right before tillage practices in spring). Six measurement series, each consisting of six point measurements, were carried out per plot. For the final field validation experiment, four plots were selected based on the collected data. One plot from each combination of non-inversion tillage/conventional tillage and the wet side (right)/dry side (left) of the field was included. Additionally, within these combinations, plots showing the highest standard deviation in maximum penetration pressures across individual measurement points were prioritized for selection to enable field validation under the most variable conditions. Figure 7 shows the experimental field, with polygons outlining the experimental plots and a color scale representing the standard deviation in maximum penetration pressure within each plot. Based on these criteria, plots B1, B3, D1, and D4 were selected.

In December 2024 (after the harvest of the maize), high-resolution penetration resistance measurements were conducted in the preselected plots. Measurement series were taken in a four-by-five grid within each plot (trajectory and task map shown in Figure 8b). Each measurement series this time consisted of seven collinear point measurements, spaced 10 cm apart. In addition, mixed soil samples were taken at 0–10 cm, 10–30 cm, and 30–60 cm depth intervals to analyze the soil moisture content. One spade of soil was collected from the central measurement point of each measurement series. Spades from the same NW-SE line and depth layer were combined into a single mixed soil sample. Soil samples were stored in plastic bags in the refrigerator, and their moisture content was determined in the laboratory by oven-drying the samples at 105 °C for 48 h. Approximately 10 cm from the robot’s measurement series, seven manual penetration resistance measurements were performed for validation, using the hand-held penetrologger (each time the same operator).

In addition to the preselected plots mentioned above, two experimental plots were selected in the spraying tracks (trajectory and task map shown in Figure 8a): one in block B and one in block C of the experimental field. In these plots, very high-resolution measurements were performed with the automated penetrometer to be able to create contour plots across the spraying track. For each spraying track plot, measurements were taken along five measurement rows perpendicular to the direction of the spraying track (Figure 9). Three of these rows were measured with the automated penetrometer, while two rows, positioned between these three rows, were measured with the hand-held penetrometer. The measurement rows of the automated penetrometer were spaced approximately 30 cm apart, with each row consisting of five measurement series. Each measurement series included seven point measurements spaced 10 cm apart. The measurement rows of the hand-held penetrometer were positioned approximately 15 cm from the rows of the automated penetrometer, with point measurements also spaced 10 cm apart. Soil sampling for soil moisture analysis was also performed in these spraying tracks following the same procedure as for the preselected plots.

### 2.5. Data Processing and Analysis

All data were processed in Python (version 3.12) and QGIS (version 3.34.10 Long Term Release). Since neither penetrometer could detect the soil surface in the cylinders under controlled conditions, the measurement data were manually shifted to ensure that the pressure was zero at the soil surface. Under these controlled conditions, the standard deviation of measurements was compared among the different measurement methods. All statistical tests were conducted at a 5% significance level. First, the assumption of normality was checked using Shapiro’s Test. Since the data were not normally distributed, a non-parametric Kruskal–Wallis test was performed to identify significant differences. To determine which methods differed, Dunn’s test was performed. Both under controlled conditions and in the field, several regression equations were calculated using the Pearson correlation coefficient as a measure of the relationship between the variables. Contour plots across the spraying tracks were created with the Plotly library [85] using the go.Contour constructor. The data were smoothed by applying a Gaussian filter using the gaussian_filter function from the SciPy library [86].

## 3. Results and Discussion

### 3.1. Calibration and Validation Under Controlled Conditions

Calibration of the hand-held (HH) penetrometer (Appendix A) was performed by comparing the measured force to that measured with a balance, resulting in the following regression equation y=1.01x−2.72 (r = 1.00). This indicates excellent alignment with the 1:1 line, confirming a correctly calibrated load cell. In contrast, applying the same calibration procedure to the automated (AU) penetrometer (Appendix A) resulted in the regression equation y=1.31x−0.48 (r = 1.0), which deviates significantly from the 1:1 line, indicating incorrect calibration. Consequently, the final data were corrected using the regression equations, assuming that the force measured by the balance represents the true force. In addition, wear on the AU penetrometer’s cone was observed, the base area was 0.78 cm^2^ instead of 1.00 cm^2^. Therefore, a correction factor was applied to derive the pressure in megapascal (often called the cone index in the literature).

In Figure 10, the mean pressure for each measurement method is plotted against depth. The three methods show similar pressure profiles with subtle differences. The HH and the robot-mounted hand-held (RM-HH) penetrometer align well, except near maximum pressure depth, possibly due to measurement variation (especially for the HH penetrometer) or the increased difficulty of inserting the HH penetrometer at higher pressures, leading to a lower insertion speed. Studies have shown that a lower insertion speed can reduce penetration resistance [87,88,89]. At lower pressures, the insertion speed of the HH device is easier to control, which may explain the absence of differences in this range. Slight deviations between the HH and the AU penetrometer are also observed. The AU penetrometer overestimates lower pressures but records a lower maximum pressure than the HH device. Since AU penetrometer measurements were taken on a different day with soil stored in non-airtight plastic bags, minor soil moisture changes may have influenced the results, as drier soil increases penetration resistance [90,91]. Soil moisture content also affects soil compressibility and stress propagation when compressing the soil into the cylinders [92,93]. Unfortunately, soil moisture content was not analyzed on different days, limiting the robustness of this interpretation. One could argue that cone wear on the AU penetrometer may have altered the cone angle and base area, leading to slight discrepancies between the HH and AU penetrometer readings. Junior et al. [87] found higher penetration pressures with a 10.98 mm^2^ cone compared to a 129.28 mm^2^ cone, and Nowatzki and Karafiath [94] showed that a decreased cone angle results in a lower cone index, especially in dense soil.

Figure 10 also includes variation bands (mean pressure ± the standard deviation). The AU penetrometer showed significantly lower standard deviations than both the HH penetrometer (Dunn’s Test: *p* = 1.55 × 10^−9^) and RM-HH penetrometer (*p* = 3.08 × 10^−3^), illustrating its repeatability. According to Carrara et al. [95] and Mome Filho et al. [96], there is a direct relationship between the reliability of the data generated by an HH penetrometer and the degree of control over the insertion speed. Herrick and Jones [97] also stated that when the penetration velocity changes (acceleration/deceleration), the soil resistance force will change accordingly, resulting in a resistance force that is not equal to the applied force. These findings indirectly support the development of automated penetrometers, which ensure a constant insertion speed and greater repeatability.

In Figure 11, the relation between the mean pressure per 1 cm depth of the AU and HH penetrometer is shown. The Pearson correlation is high (r = 0.99), indicating a strong relationship between their values, although deviations from the 1:1 line are observed. As previously described, this deviation is also evident in Figure 10; at the tail ends, the AU penetrometer curve shifts to the right of the HH penetrometer curve, while in the center, it shifts to the left. Scholz et al. [81] reported a similar regression slope (y=1.08x+0.11, r = 0.96) for loamy sand soils under laboratory conditions.

### 3.2. Field Validation

The mixed soil samples from the preselected and spraying track plots were analyzed for soil moisture content (Table 1). The mean soil moisture content exhibits low variation, ranging from 20.96% to 16.92% across different plots and soil depths. Differences between the plots are small, with a slight tendency for moisture content to decrease with increasing depth. Since the penetration resistance measurements were conducted after a long period of rain, it can be assumed that the soil moisture content was near field capacity, which is considered optimal for measuring soil penetration resistance. These soil moisture content results are less critical, as only very limited comparisons will be made between different plots. Only the two penetrometers will be compared, and since these measurements were performed next to each other (and thus under the same soil moisture conditions), soil moisture content is assumed to have only a limited effect on the validation results in the field.

In Figure 12, the mean pressure measured with the HH penetrometer is plotted against that of the AU penetrometer for the preselected plots. This mean pressure was calculated from a series of seven point measurements and averaged over 5 cm depth intervals. A linear regression yielded a Pearson correlation coefficient of r = 0.89 across all plots, with coefficients of 0.85, 0.90, 0.93, and 0.85 for plots B1, B3, D1, and D4, respectively, indicating a good correlation in each case. Scholz et al. [81] reported a mean coefficient of determination of R^2^ = 0.90 (r = 0.95) in their field validation, with regression slopes ranging from 0.91 to 1.12. However, they measured pressures above 3 MPa at only one field location, while our study recorded values up to 8 MPa. Additionally, our experimental field was highly heterogeneous, with tire tracks from harvest machinery and maize stubbles affecting the measurements. The linear regression equation of Mome Filho et al. [96], y=1.18x−0.16 (r = 0.82), indicated that penetration resistance values in manual mode were generally higher than in automated mode, which they attributed to inconsistent insertion speed and varying rod-soil contact. In our study, however, the linear regression showed a slope of 0.85, with the AU penetrometer generally recording higher mean pressures in the high-pressure range. In the low-pressure range, data align more closely with the 1:1 line, though the HH penetrometer tends to measure slightly higher pressures than the AU penetrometer. A similar explanation as before may apply; as penetration resistance increases, the insertion speed of the HH penetrometer tends to decrease, resulting in lower resistance values. Conversely, operators often increase insertion speed in low-resistance soil layers, especially after penetrating a more compacted layer, leading to a slight overestimation of penetration resistance. The AU penetrometer overcomes these limitations by maintaining a constant insertion speed, improving measurement reliability. However, this explanation should be approached with caution, as the regression slope may also be influenced by data variation and cone wear on the AU device. Data dispersion was lower at low pressures but increased at higher pressures, similar to the observations of Scholz et al. [81] and Mome Filho et al. [96], who found lower dispersion below 2 MPa. They attributed this to the difficulty of maintaining a constant speed and perpendicular rod position with the HH penetrometer as penetration resistance increases. This suggests that the AU penetrometer may be more reliable, especially in compacted soils.

Figure 13 presents pressure profile plots for all twenty measurement series in plot D1. These plots clearly show that both penetrometers produce similar pressure profiles. The plow pan is located at a depth of 30–40 cm. In some subplots, a slight vertical shift between the HH and AU penetrometer profiles is noticeable. This may be due to the less accurate depth estimation of the HH penetrometer, which relies on an ultrasonic sensor with a ground reference plate. If the reference plate is not positioned horizontally at the soil surface (reference level) or if the HH penetrometer is not perfectly level during insertion, slight errors in the depth measurement could occur, potentially affecting the correlation between both penetrometers. In contrast, the AU penetrometer uses the position of the landing legs as the zero-depth reference, ensuring stable depth estimation. Schmittmann and Lammers [98] also conducted a field validation of their tractor-mounted automated penetrometer. While they did not report correlation coefficients, their profile plots indicate good alignment of the mean pressure in the upper 30 cm. However, differences of 0 to 1 MPa were observed at depths beyond 30 cm.

Figure 14, similar to Figure 12, was created for the measurements taken in the spraying tracks. For both the AU and HH penetrometers, the points at the same position (parallel offset) along the measurement rows (Figure 9) were averaged. Overall, lower Pearson correlations were observed compared to the preselected plots in Figure 12, with a general correlation of r = 0.75. This is unsurprising since the mean was calculated from only two (HH penetrometer) or three (AU penetrometer) point measurements instead of seven. In addition, the spraying tracks are likely characterized by significant variation over short distances. These findings emphasize the importance of conducting sufficient replicate measurements, particularly in fields with high spatial variability, and highlight the need for efficient penetration resistance measurements. A better correlation was found for the spraying track in block C (r = 0.80) compared to block B (r = 0.64).

Figure 15 shows two contour plots from a cross-section through the spraying tracks in block C. Measurement row one of the HH and row two of the AU penetrometer were selected (Figure 9), as these two rows had the highest correlation (r = 0.78). Correlations were calculated between individual point measurements with the same parallel offset relative to the spraying track. The red boxes indicate the locations of the spraying tracks. The images align well, displaying a similar gradient in vertical and lateral directions. Once again, the plow pan is visible between 30 and 40 cm depth, with both penetrometers detecting it at approximately the same depth. Notably, the long-term presence of the spraying track does not appear to have a clear impact. A meta-analysis by Obour and Ugarte [99] found that multiple wheel passes primarily increase the penetration resistance in the 0–30 cm soil layer. However, Schjønning and Rasmussen [100] identified compaction effects extending to at least 64 cm in various wheeling treatments using cone penetration measurements. It is important to consider that, according to Håkansson and Lipiec [101], a penetration resistance threshold of 3 MPa is regarded as a critical limit for plant growth. Based on this criterion, our findings suggest that only the top 10–15 cm of soil would allow for unrestricted plant growth. However, no tillage practices, which could loosen the soil, were performed after harvest.

Figure 16 presents two contour plots from block B, comparing row one of the HH penetrometer with row one of the AU penetrometer. The correlation here was somewhat lower (r = 0.61), but signs of subsoil compaction were visible. High penetration resistance values were recorded at the spraying track locations, suggesting that their long-term presence may have induced subsoil compaction. This highlights the risk of serious soil degradation because of frequent wheel passes. Naderi-Boldaji et al. [102] generally found a linear relationship between the number of wheel passes and penetration resistance in the topsoil and upper subsoil. Similarly, Botta et al. [103] observed that penetration resistance increased with the number of wheel passes, up to a depth of 60 cm. By examining the legends of Figure 15 and Figure 16, it is evident that the penetration resistance at the location of the plow pan of the spraying track in block C was higher than in block B. This could be attributed to the wetter soil conditions throughout the season in block C (although no significant differences in soil moisture content were observed during the measuring period), which led to more severe compaction. In block C, no signs of the spraying track are observed, while in block B, signs are present. Chamen et al. [104] suggested that a plow pan could provide mechanical protection to the subsoil, which may explain the absence of subsoil compaction in block C.

### 3.3. Study Limitations and Future Perspectives

This study focused on a sandy loam soil under moist field conditions (at field capacity). Future research should extend the findings to different soil types and moisture levels. Further investigation into the effects of insertion speed, cone base area, and cone angle could help standardize penetration resistance measurements and clarify differences between various measuring instruments. Given the challenges of soil heterogeneity in arable fields, validation in more homogeneous conditions, such as permanent grassland, may be beneficial. Soil core collection for bulk density determination at high resolution was not included in this research due to the labor-intensive nature of the process. However, determining this soil property could have helped in assessing which penetrometer most accurately reflected the current soil compaction state. Future research should place more emphasis on the relationship between penetration resistance and bulk density, exploring alternative methods for determining bulk density, such as estimating it from auger samples. While using an autonomous robot for penetration resistance measurements is a major step forward, optimizing sampling strategies remains crucial to minimize the number of measurements while maximizing information for effective soil compaction mapping.

## 4. Conclusions

Our automated penetrometer demonstrated a strong correlation with the hand-held penetrometer under controlled conditions, producing similar pressure profiles. Additionally, it showed the lowest mean standard deviation between different point measurements, indicating high repeatability. In field conditions, the correlation was somewhat lower, reflecting the high variability of field measurements and the complexity of validating penetrometers in the field. Under both controlled and field conditions, minor discrepancies between the automated and the hand-held penetrometer were observed, possibly due to inconsistent insertion speed of the hand-held penetrometer. This suggests that the hand-held penetrometer may be less accurate, particularly in soils with high penetration resistance, an issue that can be mitigated by using the automated penetrometer, which ensures consistent insertion speed. Furthermore, the pressure profile plots revealed less accurate depth estimation with the hand-held device, which uses an ultrasonic sensor. The AU penetrometer elegantly addresses this issue by employing the landing legs as a zero-depth reference. Both penetrometers successfully identified the plow pan at a depth of 30–40 cm but were unable to clearly show the effect of the long-term presence of spraying tracks. In conclusion, the developed automated penetrometer paves the way for efficient and reliable soil compaction mapping, serving as a first step toward precision subsoiling and other targeted remediation measures.

## Figures and Tables

**Figure 1 sensors-25-01919-f001:**
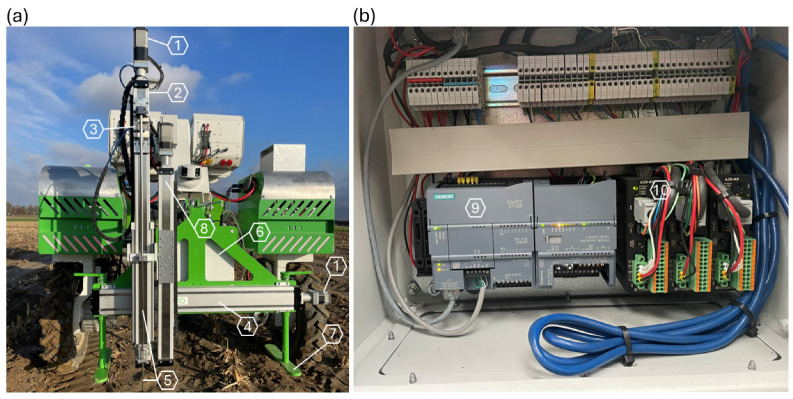
(**a**) Penetrometer hardware components: 1—stepper motor; 2—vertical linear actuator; 3—load cell; 4—horizontal linear actuator; 5—penetration rod; 6—control panel; 7—landing legs; 8—vertical linear actuator for optional soil moisture sensor. (**b**) The inside of the control panel with its components: 9—programmable logic controller; 10—motor driver.

**Figure 2 sensors-25-01919-f002:**
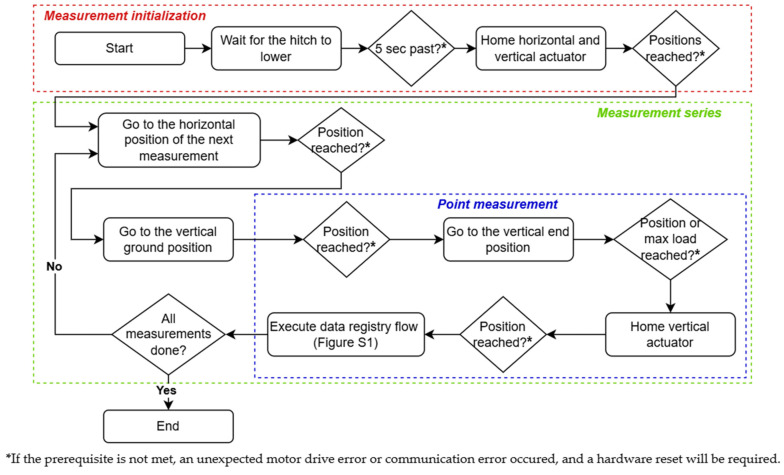
Procedure for a complete penetration resistance measurement series with the automated penetrometer.

**Figure 3 sensors-25-01919-f003:**
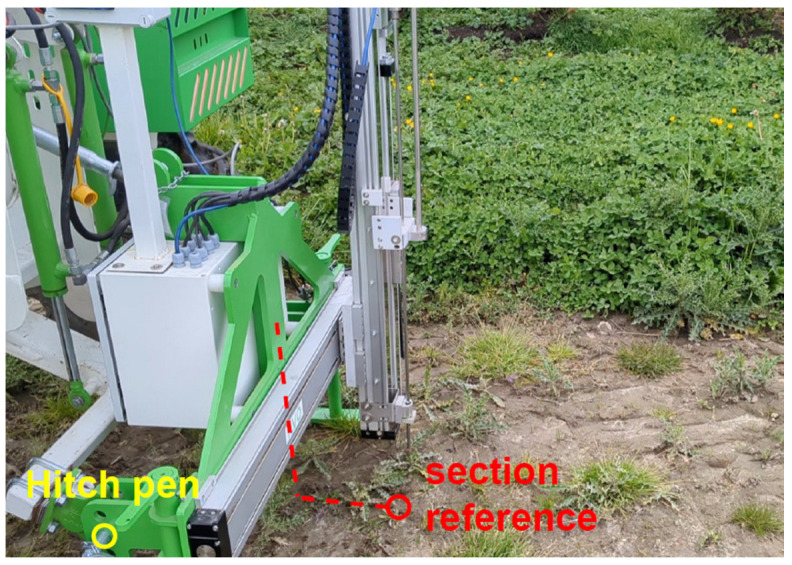
The transformation chain from the hitch hinge of the penetrometer to the section reference. For every horizontal offset position, the exact RTK-GNSS coordinates can be calculated [82] (under review).

**Figure 4 sensors-25-01919-f004:**
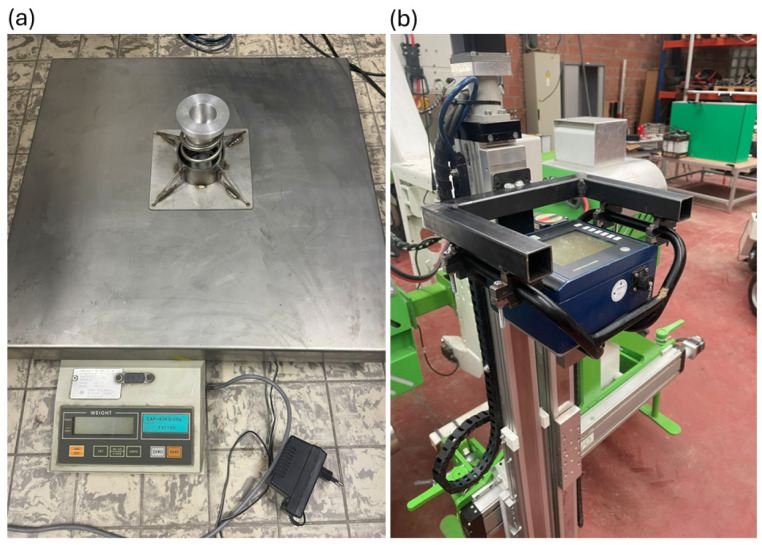
(**a**) The spring in a steel casing used for the load cell calibration of the penetrometers. (**b**) The hand-held penetrometer, mounted in a clamp on the robot’s linear actuator.

**Figure 5 sensors-25-01919-f005:**
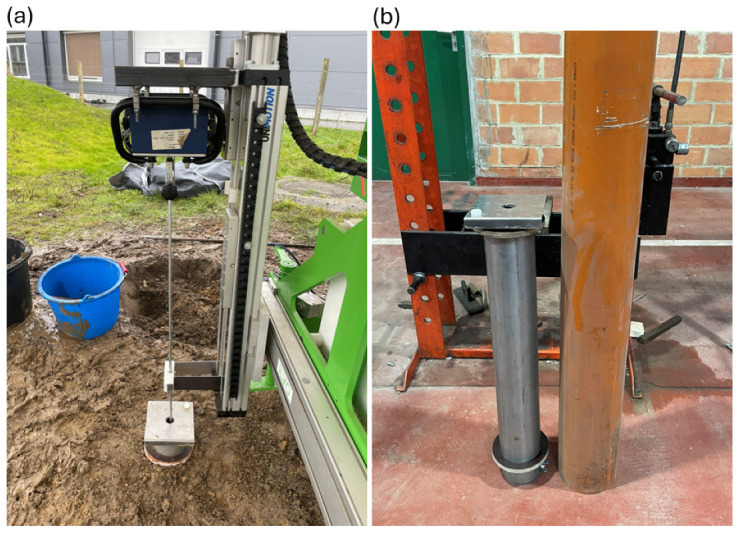
(**a**) The robot-mounted hand-held penetrometer performing a point measurement in a stabilized cylinder filled with sandy loam soil. Measurements were also taken using the manually operated hand-held penetrometer and the automated penetrometer. (**b**) The cylinder with stabilizing rings (**left**) was placed in a rain pipe (**right**), which was buried in the soil.

**Figure 6 sensors-25-01919-f006:**
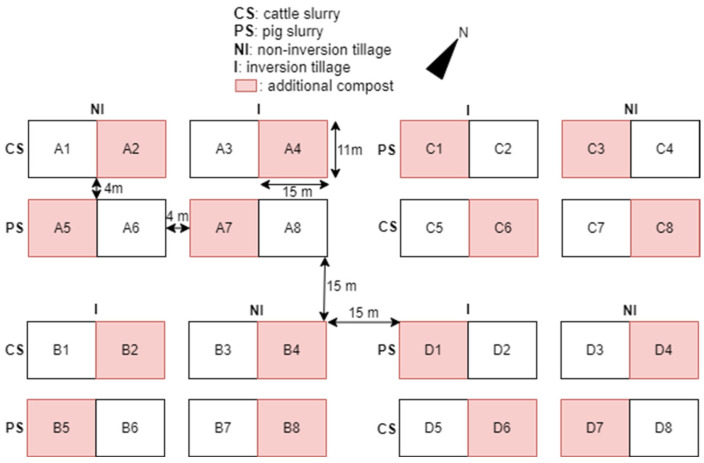
Experimental design of the multi-year BOPACT field trial.

**Figure 7 sensors-25-01919-f007:**
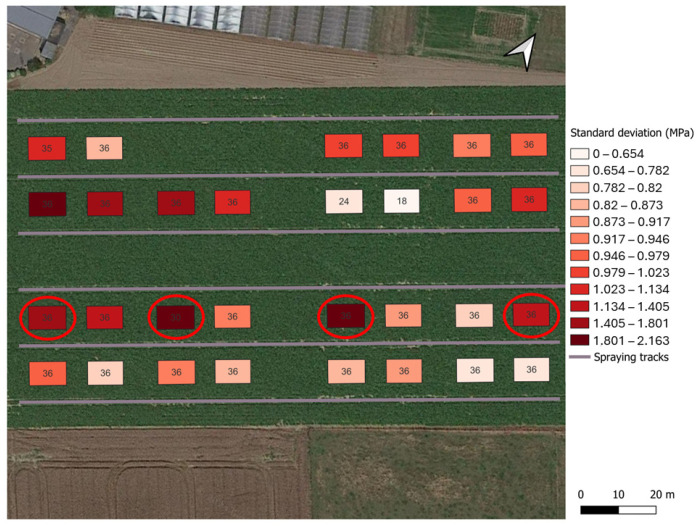
A visualization of the standard deviation in maximum penetration pressures across individual point measurements within each plot (color scale). The selected plots for the field validation are circled in red. Inside the polygons, the number of individual point measurements within each plot is indicated. Normally, 36 point measurements should be conducted in each plot. However, at the time of measurement, there were other static measurement devices present in the field, not allowing measuring at those locations.

**Figure 8 sensors-25-01919-f008:**
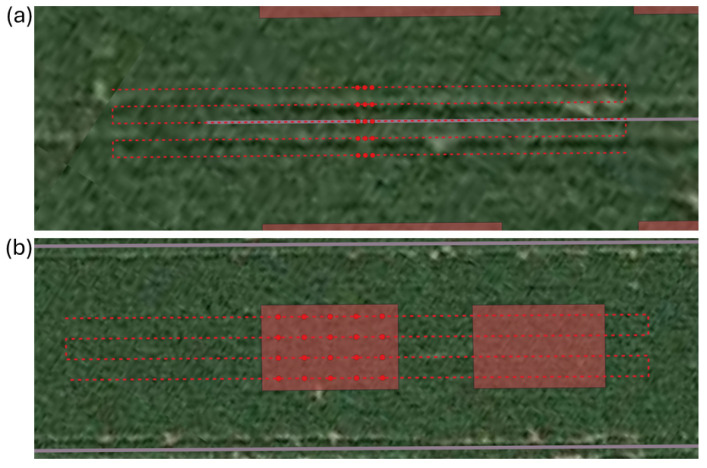
(**a**) A trajectory and a task map of a spraying track plot. (**b**) A trajectory and a task map of a preselected plot.

**Figure 9 sensors-25-01919-f009:**
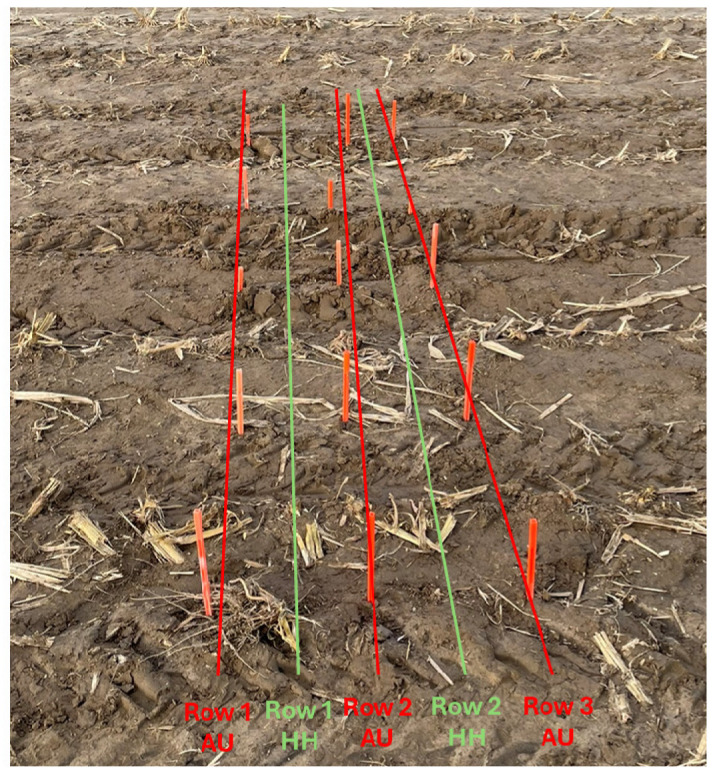
For each spraying track plot of the experimental field, penetration resistance measurements were taken along five measurement rows perpendicular to the direction of the spraying track. Three measurement rows from the automated (AU) penetrometer (red), and two measurement rows from the hand-held (HH) penetrometer (green) positioned between them. The orange rods indicate the location where the robot stopped to perform a measurement series consisting of seven point measurements spaced 10 cm apart.

**Figure 10 sensors-25-01919-f010:**
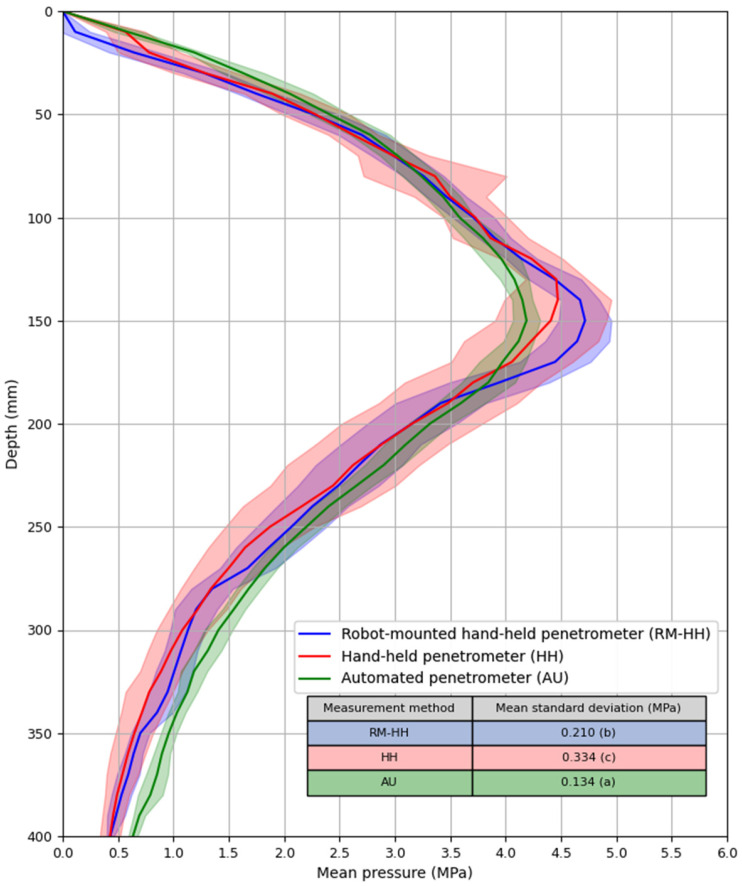
Mean pressure ± standard deviation (MPa) as a function of depth (mm) for different measurement methods. Ten replicate measurements were performed in cylinders filled with sandy loam soil with each measuring method. RM-HH: hand-held penetrometer that was clamped onto the autonomous robot’s linear actuator. HH: manually operated hand-held penetrometer. AU: automated penetrometer developed as an implement for the autonomous robot.

**Figure 11 sensors-25-01919-f011:**
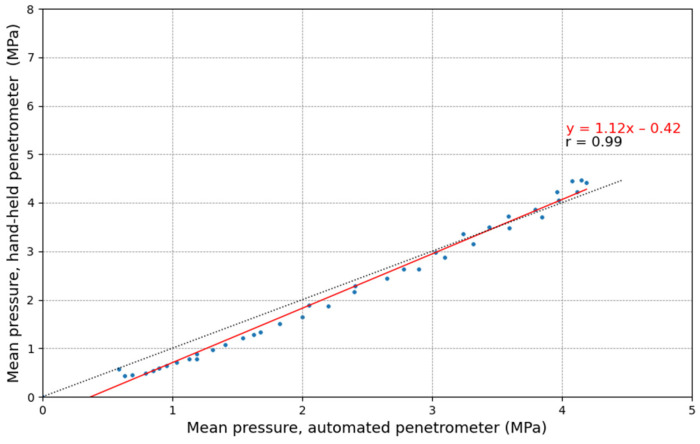
The linear relationship between the mean pressure (MPa) measured with the automated (AU) and the hand-held (HH) penetrometers under controlled conditions. The mean was calculated from ten replicate measurements in cylinders filled with sandy loam soil and averaged over 1 cm depth intervals. The correlation coefficient r in the figure corresponds to the Pearson correlation.

**Figure 12 sensors-25-01919-f012:**
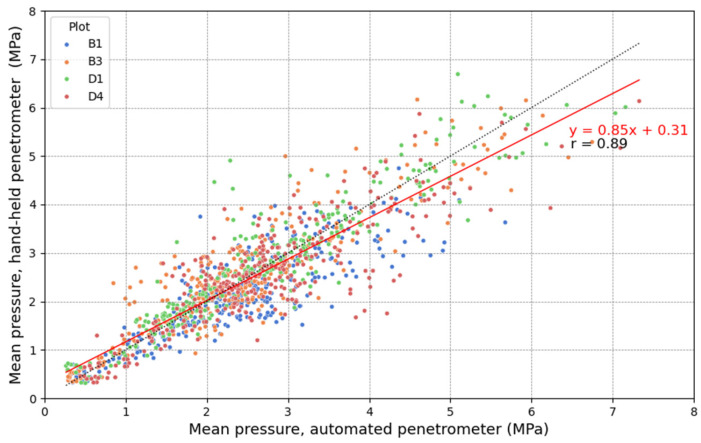
The linear relationship between the mean pressure (MPa) measured in the preselected plots with the automated (AU) and the hand-held (HH) penetrometers. The mean pressure was calculated from a series of seven point measurements and averaged over 5 cm depth intervals. The legend shows the labels of the preselected plots in the experimental field. The correlation coefficient r in the figure corresponds to the Pearson correlation.

**Figure 13 sensors-25-01919-f013:**
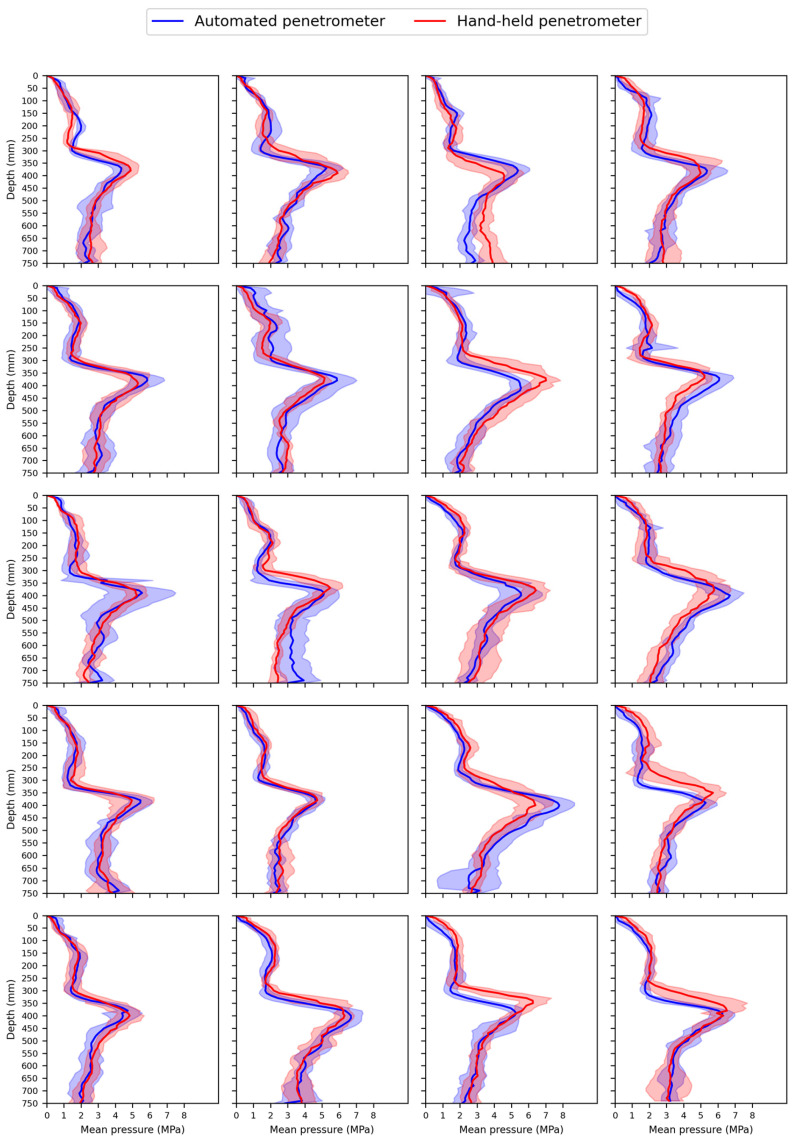
Profile plots of the mean pressure (MPa) measured with the hand-held (HH) and automated (AU) penetrometer as a function of depth (mm) for each measurement series in plot D1 of the experimental field. Each preselected plot contained a total of twenty measurement locations. At each location, a measurement series consisting of seven point measurements was performed.

**Figure 14 sensors-25-01919-f014:**
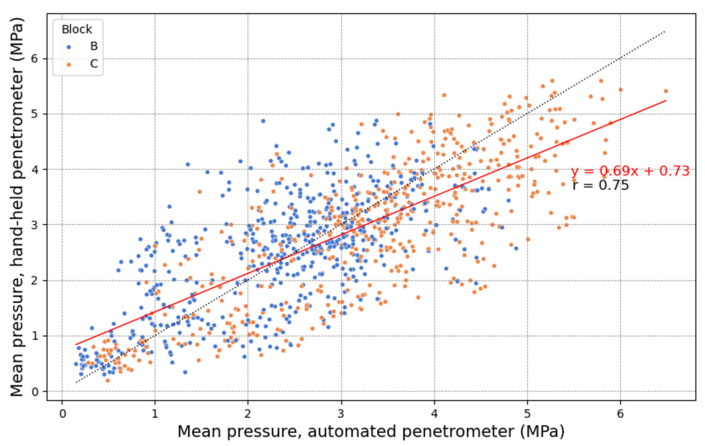
The linear relationship between the mean pressure (MPa) measured in the spraying track with the automated (AU) and the hand-held (HH) penetrometers. The measurements were taken along measurement rows perpendicular to the direction of the spraying track. Three measurement rows from the AU penetrometer, and two measurement rows from the HH penetrometer positioned between them. The mean pressure was calculated from a series of two (HH penetrometer) or three (AU penetrometer) point measurements with the same parallel offset relative to the spraying track and averaged over 5 cm depth intervals. The legend shows the labels of the block in the experimental field where the spraying track plot was located. The correlation coefficient r in the figure corresponds to the Pearson correlation.

**Figure 15 sensors-25-01919-f015:**
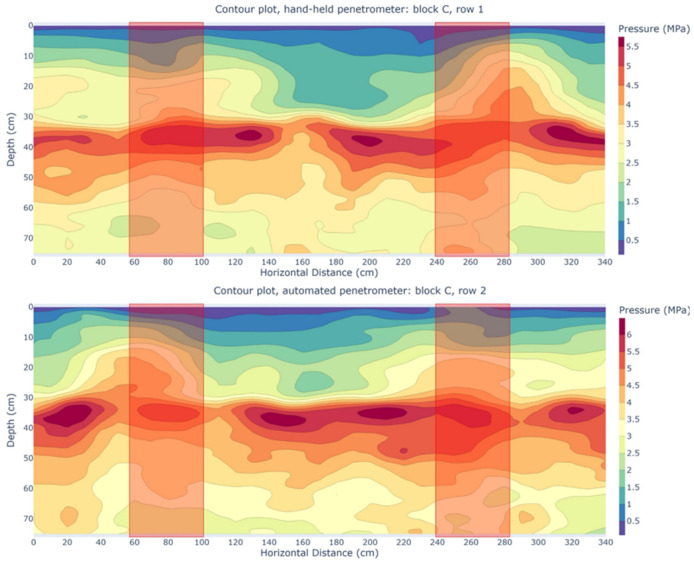
Contour plots of pressure (MPa) as a function of depth (cm) and parallel offset (cm) from measurement rows perpendicular to a spraying track in block C of the experimental field. Rows measured with the hand-held (HH) penetrometer alternated with rows measured with the automated (AU) penetrometer. Upper: contour plot of the first row measured with the HH penetrometer. Lower: contour plot of the second row measured with the AU penetrometer. The red boxes indicate the locations of the spraying tracks.

**Figure 16 sensors-25-01919-f016:**
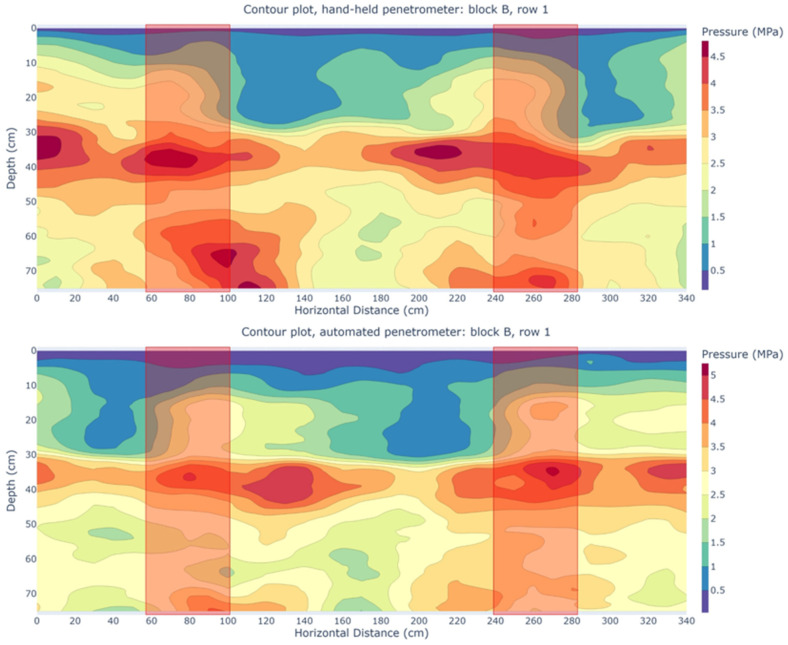
Contour plots of pressure (MPa) as a function of depth (mm) parallel offset (cm) from measurement rows perpendicular to a spraying track in block B of the experimental field. Rows measured with the hand-held (HH) penetrometer alternated with rows measured with the automated (AU) penetrometer. Upper: contour plot of the first row measured with the HH penetrometer. Lower: contour plot of the first row measured with the AU penetrometer. The red boxes indicate the locations of the spraying tracks.

**Table 1 sensors-25-01919-t001:** Mean (%) and standard deviation (%) of the gravimetric soil moisture content in the preselected plots and spraying track plots for the 0–10 cm, 10–30 cm, and 30–60 cm depth intervals.

Plot	B1	B3	D1	D4	Spraying Track B	Spraying Track C
Depth (cm)	Mean Gravimetric Soil Moisture Content (%) ± Standard Deviation (%)					
**0–10**	19.20 ± 0.15	20.96 ± 0.31	18.43 ± 0.25	20.63 ± 0.42	19.52 ± 0.09	21.94 ± 0.36
**10–30**	19.03 ± 0.36	17.91 ± 0.22	18.88 ± 0.58	17.83 ± 0.36	18.87 ± 0.46	17.63 ± 0.14
**30–60**	18.52 ± 0.41	18.02 ± 0.28	17.13 ± 0.23	17.48 ± 0.22	17.88 ± 0.07	16.92 ± 0.26

## Data Availability

Data will be made available upon request.

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
