# Peer review of "Development and In-Field Validation of an Autonomous Soil Mechanical Resistance Sensor"

_sensors, 2025, doi:10.3390/s25061919_

Round 1

Reviewer 1 Report

Comments and Suggestions for Authors

The manuscript presents an innovative piece of equipment in a very relevant research area. Currently, soil compaction monitoring using penetrometers presents challenges that this research sought to solve more efficiently.
However, some points that could be improved in the paper are shown below:
In the abstract, lines 29-31 need to review this statement, as the research did not address deformation stresses and soil stress history. Only if more analyses of density and porosity in depth were performed could infer the effect of plow pans on reducing the impact of spraying tracks.
The first paragraph of the Introduction is excessively long. I suggest summarizing the ideas in lines 41-49 in less space and avoiding repeating references between sentences.
In lines 52-61, important information on soil stress monitoring is presented, but the research did not perform these analyses. Remove them to make the Introduction more precise, direct, and succinct.
The lines highlighted in green in the Introduction can be reduced and summarized in a few sentences; they are a basis already addressed in the first paragraph. I suggest removing them from the manuscript to make the Introduction more direct.

The paragraphs between lines 98-122 can be reduced to one clearer and more direct paragraph, as they repeat concepts.

The research mentions the importance of standardizing measurements and equipment for assessing soil compaction (L146-155), but the methodology failed to relate the system's development to the standard's basis, mainly in relation to the cone size and penetration speed.

The research objectives can be summarized in normal text format, avoiding using personal language in the manuscript. I suggest traditionally rewriting them.

In the Methodology (L211-213), why were these rod and cone dimensions used? Shouldn't they be by the ASABE 313.3 penetrometry standard? Justify the changes.

L235-237 It must be highlighted that this movement is on the horizontal axis. Is that right?
L239-244 Does the ASABE standard not recommend 30 mm/s? Why was the programming done for 20 mm/s? Describe and justify these modifications. This strategy can improve the quality of the measurements, but it is necessary to cite references or justify this choice.
L242 - Why was the force determined at a maximum of 784.9 N? What happens if it exceeds this value? Explain better.
Figure 2 needs to be revised because if some points in the routine have the answer "no," what happens to the process? See the example of "position Reached?" These steps need to be revised due to the signal's positive and negative feedback.
Figure 2 also lacks the procedure for storing the collection data. Should we save it and move on to the next collection?
In the methodology between lines 290-307, it is not clear why the portable penetrometer was calibrated on cylinders. It is also essential to highlight the effect of the robot on its penetration performance since this equipment is usually operated manually by humans, with more force variations. Wouldn't it have been interesting to have also compared it this way? Review and improve this justification for the manuscript.

The soil and its characteristics in the tests were not fully detailed, especially moisture at each depth. This is essential for comparing the systems and replicating the results in other conditions.

Figures 7 and 8 should be cited immediately after they appear in the text.

Could the methodology described between lines 338-373 be illustrated with a figure or flowchart? This would make the test format more transparent.

Figures 8 and 9 could be combined as details of the collections in plots.

L406-448 This paragraph is excessively long. I suggest subdividing and summarizing its contents. Some information regarding the methodology can be removed. Once again, it is important to relate the soil water content during the tests with the results obtained.

Figure 10 shows the three collection methods, which the methodology must describe in detail. The same applies to statistical analyses that are not fully described in the method. Review and reorganize the tests and analyses in the methodology in the same order they are presented in the results.
L461-466 I suggest that this table be integrated into the main document and cited in the methodology. Consider creating a graph or figure of these in-depth analyses to help explain the cone index.
L488-506 It is crucial to discuss the differences between the equipment developed and the traditional system in terms of improving collections. A high r2 does not mean your equipment has improved the existing collection project. An approach to improving the collection procedure should be taken because, from my point of view, the equipment developed by the research may present more representative soil data and, even so, have a low r2 compared to the traditional method. Verify and expand on this issue.
How were the maps in Figure 15 prepared? Describe the methodology. Which data interpolation method was used? Kriging? Also, describe the software and specific library used to construct the figures.
Figures 15 and 16 lacked further exploration with citations about the quality of the methods in determining subsoil stresses.
The conclusion can be improved by highlighting the qualities of the equipment developed and its ability to accurately determine soil compaction in relation to traditional methods. Also, highlight the changes made to the penetrometer construction standard and their advantages.

Congratulations on the research! I found it very interesting. It will open doors for new research into soil compaction and more agile and low-cost field assessment methods.

Author Response

We are grateful for the relevant and insightful feedback you provided, which has elevated the quality of this paper. We have made every effort to adjust the paper according to your remarks and suggestions.

Comment 1 : "In the abstract, lines 29-31 need to review this statement, as the research did not address deformation stresses and soil stress history. Only if more analyses of density and porosity in depth were performed could infer the effect of plow pans on reducing the impact of spraying tracks."

Response 1: Agree. We did only mildly suggest it, but it indeed cannot be further substantiated with our research results. Therefore, we have removed these lines from the abstract and rewritten it to reflect the inability to always detect an effect of the long-term presence of spraying tracks (L29-31, see the attached revised document): ”Both penetrometers successfully identified the plow pan at a depth of 30-40 cm, but were unable to clearly show the effect of the long-term presence of spraying tracks.”

Comment 2-5: “The first paragraph of the Introduction is excessively long. I suggest summarizing the ideas in lines 41-49 in less space and avoiding repeating references between sentences.
In lines 52-61, important information on soil stress monitoring is presented, but the research did not perform these analyses. Remove them to make the Introduction more precise, direct, and succinct.
The lines highlighted in green in the Introduction can be reduced and summarized in a few sentences; they are a basis already addressed in the first paragraph. I suggest removing them from the manuscript to make the Introduction more direct.

The paragraphs between lines 98-122 can be reduced to one clearer and more direct paragraph, as they repeat concepts.”

Response 2-5: We agree with this comment and acknowledge that, in general, the introduction was too detailed. We aimed to provide a comprehensive overview of the highly relevant issue of soil compaction, but in doing so, we lost focus on the core objective of this research paper. Therefore, we:

  • Summarized L41-49 into L41-44 
  • Removed L52-59 and integrated L60-61 into L49-51
  • Summarized L68-89 into L57-69
  • Condensed the two paragraphs between L98-122 into one shorter paragraph (L76-91)

Comment 6: “The research mentions the importance of standardizing measurements and equipment for assessing soil compaction (L146-155), but the methodology failed to relate the system's development to the standard's basis, mainly in relation to the cone size and penetration speed.”

Response 6: We agree that this might be somewhat confusing. We stated that there is a standard that is sometimes followed by researchers but is not universally accepted or utilized. For the design of our penetrometer, we aimed to ensure the comparability with the hand-held penetrometer, which is frequently used at our research institute as well as throughout Belgium and the rest of Europe. We have added L180-183 in the methodology section to further justify our choice: “Since the ASABE standard is not widely accepted and to ensure comparability with the recommended cone for general fieldwork with the hand-held Eijkelkamp penetrometer (06.15.SA), which is commonly used in Belgium, a penetration cone with a 60° angle and a base area of 1 cm² was chosen.”

Comment 7: “The research objectives can be summarized in normal text format, avoiding using personal language in the manuscript. I suggest traditionally rewriting them.”

Response 7: Agree. Accordingly, we have rewritten the objectives in standard text (L149-152): “This paper aims to describe the in-house developed, autonomous robot-mounted automated penetrometer for high-resolution soil compaction mapping and validate this penetrometer under both controlled and field conditions by comparing it with manual penetrometer measurements.”

Comment 8: In the Methodology (L211-213), why were these rod and cone dimensions used? Shouldn't they be by the ASABE 313.3 penetrometry standard? Justify the changes.

Response 8: We agree that this was unclear. Accordingly, as mentioned above, we added L180-183 to justify our choice. The recommended cone for general fieldwork has an angle of 60° and a base area of 1 cm², as specified in the user manual of the Eijkelkamp penetrometer (see https://www.royaleijkelkamp.com/media/nrwjyah3/m-0615sae-penetrologger.pdf). For your information, different cones can be used with the automated penetrometer.

Comment 9: “L235-237 It must be highlighted that this movement is on the horizontal axis. Is that right?”

Response 9: We understand the confusion, but the movement is vertical. We have adapted L205-207 to clearly indicate the direction of the movement: ”To conduct a point measurement, the penetrometer's cone tip moves vertically to the ground position (6550 steps = 3.275 cm) at a speed of 15 cm/s.”

Comment 10: “L239-244 Does the ASABE standard not recommend 30 mm/s? Why was the programming done for 20 mm/s? Describe and justify these modifications. This strategy can improve the quality of the measurements, but it is necessary to cite references or justify this choice.”

Response 10: We again made that choice to ensure comparability with the hand-held penetetrometer of Eijkelkamp, which is usually operated at an insertion speed of 2 cm/s, as recommended by the device's operation manual. We further justified this in L209-212: ”The penetrometer rod is inserted into the soil at a velocity of 2 cm/s, following the recommended operation speed of the hand-held Eijkelkamp penetrometer, as specified in the device's manual [83].”

Comment 11: “L242 - Why was the force determined at a maximum of 784.9 N? What happens if it exceeds this value? Explain better.”

Response 11: That might indeed be a little confusing. The raw data of the load cell is provided in weight units (kg), and a safety value of 80 kg = 748.9N was set. This maximum value is set to prevent damage to the penetrometer, as it represents the mechanical limit of the vertical linear actuator. We have made some adjustments to L213-216 for clarity: ”This process continues until the rod reaches the vertical end position (ground position + 75 cm) or the measured force exceeds a preset maximum value of 80 kg (=784.9 N). This maximum value is set to prevent damage to the penetrometer, as it represents the mechanical limit of the vertical linear actuator.”

Comment 12: “Figure 2 needs to be revised because if some points in the routine have the answer "no", what happens to the process? See the example of "position Reached?" These steps need to be revised due to the signal's positive and negative feedback.
Figure 2 also lacks the procedure for storing the collection data. Should we save it and move on to the next collection?”

Response 12: Agree, this needs further explanation. We have adjusted Figure 2 by adding the element “Execute data registry flow” which indicates the link with section 2.2.3. This flow can also be consulted in the supplementary material. So, while measuring, the data is stored on the SD card of the PLC, and after the point measurement is completed, this data is extracted via the data registry flow. We also added an asterisk to the flowchart to explain what happens if the answer to the prerequisite questions is “no”. In that case, an unexpected motor drive error or communication error has occurred.

Comment 13: “In the methodology between lines 290-307, it is not clear why the portable penetrometer was calibrated on cylinders. It is also essential to highlight the effect of the robot on its penetration performance since this equipment is usually operated manually by humans, with more force variations. Wouldn't it have been interesting to have also compared it this way? Review and improve this justification for the manuscript.”

Response 13: We acknowledge that we did not justify the choice of the cylinders as a validation medium, so we have added L292-293: ”This procedure allowed for multiple replicate measurements in a standardized manner.” However, the hand-held penetrometer was calibrated using a balance, following the same procedure as the automated penetrometer, which actually only involves a calibration of the load cells of both devices (excluding factors like manual operation). Then we conducted a validation experiment in the cylinders, testing the hand-held device both in automated mode (clamped onto the linear actuator) and manual mode (hence, including the human factor). Small adjustments were made to the text between L263 and L301 for clarity, and the figures in that section were slightly revised.

Comment 14: “The soil and its characteristics in the tests were not fully detailed, especially moisture at each depth. This is essential for comparing the systems and replicating the results in other conditions.”

Response 14: We understand this comment. However,  for all three measuring methods used in the experiment with the soil cylinders, the same soil was utilized, which was carefully mixed and crumbled before being used to fill the cylinders. Hence, the soil moisture content remained constant as a function of depth in these cylinders. Given this, we did not perform soil moisture content analysis on this soil, which indeed could be a limitation regarding the interpretation, as we now mention in L424-425: “Unfortunately, we did not analyze the soil moisture content on the different days, which limits the robustness of this interpretation.” However, the use of non-airtight plastic bags likely resulted in only very small differences in soil moisture.

Comment 15: “Figures 7 and 8 should be cited immediately after they appear in the text.”

Response 15: As suggested, we have moved Figures 7 and 8 earlier in the text to L333 and L354 respectively.

Comment 16:”Could the methodology described between lines 338-373 be illustrated with a figure or flowchart? This would make the test format more transparent.”

Response 16: We appreciate your suggestion, however, we believe that there are already many figures included in this paper. Therefore, we hope that the approach is clear from the text and the figures that are already presented. We further reviewed the text for transparency and repeatability of the methodology.

Comment 17: “Figures 8 and 9 could be combined as details of the collections in plots.”

Response 17: We fully understand this suggestion, but combining the figures results in a small combined figure, making the details less visible.

Comment 18: “L406-448 This paragraph is excessively long. I suggest subdividing and summarizing its contents. Some information regarding the methodology can be removed. Once again, it is important to relate the soil water content during the tests with the results obtained.”

Response 18: Agree. We have subdivided that paragraph into three separate ones (L408-453) and made some small improvements to the text for better readability. Regarding the soil moisture content, the three methods were applied on the same homogenized soil as mentioned in response 14.

Comment 19: “Figure 10 shows the three collection methods, which the methodology must describe in detail. The same applies to statistical analyses that are not fully described in the method. Review and reorganize the tests and analyses in the methodology in the same order they are presented in the results.”

Response 19: We agree, this should now be clear in the methodology, as we have added section 2.5 (L372-382).

Comment 20: “L461-466 I suggest that this table be integrated into the main document and cited in the methodology. Consider creating a graph or figure of these in-depth analyses to help explain the cone index.”

Response 20: Agree. However, we (wrongly) considered this table less critical, since the soil moisture variations were small. In addition, the focus was on comparing the measurements of the AU penetrometer to those of the HH penetrometer, where measurements were taken at the same location (and thus at the same moisture content). We did not primarily intend to compare the results across different plots. For completeness, we have included Table 1 in the results section and further elaborated on this in L460-471.

Comment 21: “L488-506 It is crucial to discuss the differences between the equipment developed and the traditional system in terms of improving collections. A high r2 does not mean your equipment has improved the existing collection project. An approach to improving the collection procedure should be taken because, from my point of view, the equipment developed by the research may present more representative soil data and, even so, have a low r2 compared to the traditional method. Verify and expand on this issue.”

Response 21: The primary goal was to demonstrate that the AU penetrometer delivers results comparable to the HH penetrometer. However, it is true that this research also provided some arguments suggesting that the AU penetrometer might be more reliable, as described throughout the results section. We have further elaborated on them in L492-494: ”In contrast, the developed AU penetrometer ensures a constant insertion rate and stable rod-soil contact, enhancing measurement reliability.” , L503-504: ”The AU penetrometer overcomes these limitations by maintaining a constant insertion speed.”, L514-517: ”This suggests that the AU penetrometer may be more reliable than the HH penetrometer, particularly in highly compacted soils, as maintaining consistent operation with the HH penetrometer becomes more challenging.”

Comment 22: “How were the maps in Figure 15 prepared? Describe the methodology. Which data interpolation method was used? Kriging? Also, describe the software and specific library used to construct the figures.”

Response 22: We agree. Accordingly, we have added section 2.5 (L391-394: ”Contour plots across the spraying tracks were created with the Plotly library [85] using the go.Contour constructor. The data were smoothed by applying a Gaussian filter using the gaussian_filter function from the SciPy library [86].)

Comment 23: “Figures 15 and 16 lacked further exploration with citations about the quality of the methods in determining subsoil stresses.”

Response 23: Agree, we further elaborated on this in L577-579: ”However, Schjønning and Rasmussen [100] identified compaction effects extending to at least 64 cm (the maximum measuring depth) in various wheeling treatments using cone penetration measurements.” and L598-601: ”Naderi-Boldaji, et al. [102] generally found a linear relationship between the penetration resistance and the number of wheel passes in the topsoil and upper subsoil. Similarly, Botta, et al. [103] observed that penetration resistance increased with the number of wheel passes in a conventional tillage system, up to a depth of 60 cm.”

Comment 24: “The conclusion can be improved by highlighting the qualities of the equipment developed and its ability to accurately determine soil compaction in relation to traditional methods. Also, highlight the changes made to the penetrometer construction standard and their advantages.”

Response 24: We agree and have adjusted the conclusion to emphasize the qualities of the developed penetrometer. L637-638: ”Additionally, it showed the lowest mean standard deviation between different point measurements, indicating high repeatability.” L642-648: ”This suggests that the hand-held penetrometer may be less accurate, particularly in soils with high penetration resistance, an issue that can be mitigated by using the automated penetrometer, which ensures consistent insertion speed. Furthermore, the pressure profile plots revealed less accurate depth estimation with the hand-held device, which uses an ultrasonic sensor. The AU penetrometer elegantly addresses this issue by employing the landing legs as a zero-depth reference.” The reason we did not choose the ASABE standard is explained in the methodology section.

Reviewer 2 Report

Comments and Suggestions for Authors

This study developed an automated soil compaction detection device, achieving automation in soil penetration resistance measurement through the design of an automated system. The research includes the design and integration of the hardware system for the automated penetration resistance device, comprising a probe, pressure sensor, and PLC controller. A laboratory calibration experiment was conducted using a soil-filled cylinder, while data acquisition and processing were implemented via Node-RED. Field trials were conducted in multiple test areas, comparing the automated device with a traditional handheld soil penetration resistance instrument. The results demonstrated that the automated measurements closely matched manual measurements while offering significantly higher efficiency. Additionally, pressure contour maps were generated to visualize soil compaction levels. The paper is interesting, and it can be published if the authors address the following comments in the revised paper and satisfy the editors and reviewers:

1. Line 110 of the introduction summarizes the main methods currently used for soil compaction measurement, including handheld penetrometers, towed sensors, and geophysical techniques (such as GPR and ERT). To better highlight the necessity of the automated penetrometer, it is essential to provide a detailed comparison of the advantages and limitations of each method.

2. In Section 2.3, soil was packed into a cylindrical steel container for the calibration of the penetrometer. Was the soil subjected to a homogenization process before calibration to ensure uniformity?

3. The field experiment only mentions the collection and measurement of mixed soil samples, but does not specify the measurement methods used (e.g., oven-drying method, TDR sensors, etc.). It is recommended that the authors provide a detailed description of the process of sampling, handling, and measurement of soil moisture content, including the number of samples, depth of sampling, and specific measurement techniques.

4. Since soil moisture is a critical factor influencing soil penetration resistance, it is recommended to include relevant literature in the Discussion section. This should provide supporting evidence and analyze how variations in soil moisture content affect soil mechanical resistance measurements.

5. If independent soil bulk density measurements were not feasible, it is recommended to explicitly discuss this limitation in the Discussion section. Additionally, analyzing its potential impact on result interpretation and conclusions would enhance the rigor of the study.

Comments on the Quality of English Language

The English could be improved to more clearly express the research.

Author Response

We are grateful for the relevant and insightful feedback you provided, which has elevated the quality of this paper. We have made every effort to adjust the paper according to your remarks and suggestions. We also reviewed the text for sentence structure to enhance readability.

Comment 1: “Line 110 of the introduction summarizes the main methods currently used for soil compaction measurement, including handheld penetrometers, towed sensors, and geophysical techniques (such as GPR and ERT). To better highlight the necessity of the automated penetrometer, it is essential to provide a detailed comparison of the advantages and limitations of each method.”

Response 1: We agree that this is an important aspect. We believe we already mentioned the main (dis)advantages of the other methods from L92 onwards, namely that traditional measurements based on soil sampling are invasive, disturb the soil, and require laboratory analysis, which is time- and resource-intensive [6,8,53,54], while geophysical techniques have had varying degrees of success [55-62]. In addition, these methods do not provide accurate profile information regarding compaction, and the results are sometimes difficult to interpret.

Since another reviewer recommended making the introduction more concise, we decided not to elaborate on every measuring method, as our aim is not to provide an exhaustive overview of current techniques. Most of these methods are also still under investigation, while vertical penetrometers are commonly used today.

Comment 2: “ In Section 2.3, soil was packed into a cylindrical steel container for the calibration of the penetrometer. Was the soil subjected to a homogenization process before calibration to ensure uniformity?”

Response 2: That is indeed important to mention. Yes, the soil was preprocessed, and we have further clarified this in L289-290: ”Before filling the cylinders, the soil was homogenized by mixing and crumbling.”

Comment 3: “The field experiment only mentions the collection and measurement of mixed soil samples, but does not specify the measurement methods used (e.g., oven-drying method, TDR sensors, etc.). It is recommended that the authors provide a detailed description of the process of sampling, handling, and measurement of soil moisture content, including the number of samples, depth of sampling, and specific measurement techniques.”

Response 3: We agree this was unclear. We further elaborated on the soil moisture analysis in L344-349 for better clarity: ”In addition, mixed soil samples were taken at 0–10 cm, 10–30 cm, and 30–60 cm depth intervals to analyze the soil moisture content. One spade of soil was collected from the central measurement point of each measurement series. Spades from the same NW-SE line and depth layer were combined into a single mixed soil sample. Soil samples were stored in plastic bags in the refrigerator, and their moisture content was determined in the laboratory by oven-drying the samples at 105°C for 48 hours.”

Comment 4: “Since soil moisture is a critical factor influencing soil penetration resistance, it is recommended to include relevant literature in the Discussion section. This should provide supporting evidence and analyze how variations in soil moisture content affect soil mechanical resistance measurements.”

Response 4: Agree. We have added Table 1 (and L460-471) with the soil moisture results of the field experiment to provide clarity. This table shows that the variation in soil moisture content was minimal, as the soil was near field capacity during the measurement period. However, the primary focus was to compare the measurements of the AU penetrometer to those of the HH penetrometer, taking measurements at the same location (and hence same moisture content). We did not intend to compare the results across different plots, which is why we considered soil moisture less critical in this research.

Comment 5: “If independent soil bulk density measurements were not feasible, it is recommended to explicitly discuss this limitation in the Discussion section. Additionally, analyzing its potential impact on result interpretation and conclusions would enhance the rigor of the study.”

Response 5: We understand this comment and have therefore highlighted it as a limitation in section 3.3: Study Limitations and Future Perspectives. L625-630:”Soil core collection for bulk density determination at high resolution was not included in this research due to the labor-intensive nature of the process. However, determining this soil property could have helped in assessing which penetrometer most accurately reflected the current soil compaction state. Future research should place more emphasis on the relationship between penetration resistance and bulk density, exploring alternative methods such as estimating bulk density from auger samples.”

Reviewer 3 Report

Comments and Suggestions for Authors

The paper presents an innovative and promising study, developing and validating an autonomous robot-mounted soil penetration resistance sensor. It offers a significant contribution to soil compaction monitoring in precision agriculture. The article is well-structured, with a sound experimental design and clear data presentation. However, the following issues warrant attention:

  1. The practical application value of this study is not sufficiently emphasized in the abstract, making it difficult for readers to quickly grasp its real-world significance. It is recommended that the authors clearly outline the study’s relevance to precision agriculture or soil monitoring to enhance its practicality and appeal.
  2. The authors do not provide data on the device’s operational power consumption, making it difficult for readers to assess its ability to operate continuously in the field. It is recommended to include power consumption details, such as energy usage and battery endurance, to better evaluate its practicality and scalability.
  3. The paper sporadically mentions factors that may affect the results (e.g., soil heterogeneity, cone wear, number of repetitions) but does not systematically summarize the study’s limitations. It is suggested that the authors consolidate these aspects in the discussion section and propose improvement strategies and future research directions to enhance the completeness and readability of the paper.

Author Response

We are grateful for the relevant and insightful feedback you provided, which has elevated the quality of this paper. We have made every effort to adjust the paper according to your remarks and suggestions.

Comment 1: “The practical application value of this study is not sufficiently emphasized in the abstract, making it difficult for readers to quickly grasp its real-world significance. It is recommended that the authors clearly outline the study’s relevance to precision agriculture or soil monitoring to enhance its practicality and appeal.”

Response 1: Agree. We have further emphasized the relevance of this development throughout the text, particularly in the abstract (L19-21:”To address these limitations an automated penetrometer was developed and integrated on an autonomous robot platform paving the way for high-resolution compaction mapping as a starting point for for precision subsoiling to remediate soil compaction.”) and conclusion (L650-652:”In conclusion, the developed automated penetrometer paves the way for efficient and reliable soil compaction mapping, serving as a first step toward precision subsoiling and other targeted remediation measures.”).

Comment 2: “The authors do not provide data on the device’s operational power consumption, making it difficult for readers to assess its ability to operate continuously in the field. It is recommended to include power consumption details, such as energy usage and battery endurance, to better evaluate its practicality and scalability.”

Response 2: Agree. We have added L161-163 to clarify this: ”The mean power consumption of the robot when performing penetrometer measurements is approximately 2.36 kW, which means that, with a battery capacity of 20 kWh, the robot can operate continuously for an estimated duration of more than 8 hours.”

Comment 3: “The paper sporadically mentions factors that may affect the results (e.g., soil heterogeneity, cone wear, number of repetitions) but does not systematically summarize the study’s limitations. It is suggested that the authors consolidate these aspects in the discussion section and propose improvement strategies and future research directions to enhance the completeness and readability of the paper.”

Response 3: Agree. Accordingly, we elaborated more on the limitations and proposed future research direction in section 3.3: Study Limitations and Future Perspectives. L619-633: ”This study focused on a sandy loam soil under moist field conditions (at field capacity). Future research should extend the findings to different soil types and moisture levels. Further investigation into the effects of insertion speed, cone base area, and cone angle could help standardize penetration resistance measurements and clarify differences between various measuring instruments. Given the challenges of soil heterogeneity in arable fields, validation in more homogeneous conditions, such as permanent grassland, may be beneficial. Soil core collection for bulk density determination at high resolution was not included in this research due to the labor-intensive nature of the process. However, determining this soil property could have helped in assessing which penetrometer most accurately reflected the current soil compaction state. Future research should place more emphasis on the relationship between penetration resistance and bulk density, exploring alternative methods such as estimating bulk density from auger samples. While using an autonomous robot for penetration resistance measurements is a major step forward, optimizing sampling strategies remains crucial to minimize the number of measurements while maximizing information for effective soil compaction mapping.” 

Round 2

Reviewer 1 Report

Comments and Suggestions for Authors

The manuscript has been modified and improved, but some points still need to be modified:

The last paragraph of the introduction could be shortened, as was done with the previous ones. I suggest subdividing the text between L92 and L152.

In the methodology, it is necessary to describe the commercial penetrometer (Eijkelkamp, ​​06.15.SA).

In the results, the contexts of the long paragraphs can be summarized and made more direct. This occurs in the lines: L408-430; L475-517;

Comments on the Quality of English Language

The English language can be improved and the text clearer. Shorten the long paragraphs to make reading more fluid.

Author Response

We would like to thank you once again for your detailed feedback, which has significantly enhanced the quality of this paper.

Comment 1: “The last paragraph of the introduction could be shortened, as was done with the previous ones. I suggest subdividing the text between L92 and L152.”

Response 1: Agree. We shortened the paragraph and divided the text into four sections (L91-142).

Comment 2: “In the methodology, it is necessary to describe the commercial penetrometer (Eijkelkamp, 06.15.SA).”

Response 2: We provided a short description of the manual penetrometer in L255-258: “The hand-held device includes a built-in datalogger and an integrated GPS, and it records the penetration resistance at every 1 cm depth interval, up to a maximum depth of 80 cm. The rod and cone specifications of the hand-held penetrometer are identical to those of the automated penetrometer.

Comment 3: “In the results, the contexts of the long paragraphs can be summarized and made more direct. This occurs in the lines: L408-430; L475-517;”

Response 3: We summarized these paragraphs, retaining only the essential parts:

  • L408-430 -> L398-418
  • L475-517 -> L463-492
